## RESEARCH ARTICLE

# Activity-dependent remodeling of muscle architecture during distinct locomotor behaviors in *Caenorhabditis elegans*

Adina Fazyl, Akash Anbu, Sabrina Kollbaum and Andrés G. Vidal-Gadea*

## ABSTRACT

Muscle structure is dynamically shaped by mechanical use, yet how distinct locomotor behaviors influence sarcomere organization remains poorly understood. In *Caenorhabditis elegans*, crawling and swimming constitute discrete gaits that differ in curvature, frequency, and mechanical load, providing a tractable model for studying activity-dependent remodeling. Using confocal imaging of phalloidin-stained body-wall myocytes, we quantified myocyte geometry, sarcomere length, and sarcomere number across anterior, mid-body, and posterior regions in animals reared exclusively under crawling or swimming conditions. Quantification and hypothesis testing used linear mixed models that accounted for repeated myocyte measurements within animals, with interaction terms testing region-specific effects of locomotor condition after interquartile range (IQR)-based outlier removal. Swimming produced characteristic remodeling of body-wall muscles. Myocytes elongated globally, while selectively thinning in the mid-body, reducing cell area by ~13% relative to crawlers. Shape metrics confirmed this shift: circularity declined at mid- and tail-regions and anisotropy increased by ~2–3 units. Sarcomere architecture exhibited parallel remodeling. Average sarcomere length shortened across the body (−0.19 μm in head, −0.35 μm in mid-body, −0.20 μm in tail), while sarcomere number increased in anterior and mid-body regions (+0.77 and +0.65 sarcomeres per myocyte). The mid-body region also showed a significant rise in sarcomere density, indicating tighter serial packing. These adaptations mirror functional compartmentalization predicted from gait kinematics and parallel fast-fiber remodeling observed in vertebrate muscles. The results indicate that *C. elegans* muscles adapt their contractile lattice to sustained mechanical demand, linking neural gait selection and mechanosensitive signaling to long-term structural plasticity. This work establishes *C. elegans* as a model for dissecting the conserved pathways that couple muscle use to cellular architecture and provides a foundation for future comparisons of healthy and diseased muscle remodeling.

KEY WORDS: *C. elegans*, Sarcomere, Muscle plasticity, Locomotion, Mechanotransduction, Activity-dependent remodeling

## INTRODUCTION

Animals adapt their movement patterns to the physical properties of their environment, and muscles must structurally accommodate these different mechanical demands. In *Caenorhabditis elegans*, locomotion alternates between two distinct gaits: crawling on solid surfaces and swimming in liquid. Complementary behavioral and genetic analyses confirmed that transitions between solid and liquid evoke distinct locomotor programs rather than a single continuum of speeds (Pierce-Shimomura et al., 2008), establishing that crawl and swim represent bona fide gaits with different mechanical characteristics. Early work from our group demonstrated that these gaits are selected through aminergic modulation, with serotonin promoting the transition from crawling to swimming, and dopamine stabilizing crawling once animals return to land (Vidal-Gadea et al., 2011).

Body wall muscles in *C. elegans* are not uniform along the anterior–posterior axis. Developmental reconstructions already indicate that adult midbody body wall myocytes contain more sarcomeres in series than cells near the head or tail, revealing a baseline regional gradient in sarcomere number that emerges during growth rather than as an acute response to use (Gieseler et al., 2017). Biomechanical and kinematic analyses further show that curvature, bending strain, and bending forces concentrate near the midbody during locomotion in both liquid and on wet agar, establishing this region as the primary site of mechanical demand during undulatory movement (Pierce-Shimomura et al., 2008; Shen et al., 2012; Fang-Yen et al., 2010). More recent work has quantified regional differences in myocyte cross-sectional area in the context of microgravity, nutritional deficiency, and disuse, again revealing that specific subsets of body wall muscles are differentially sensitive to environmental challenge (Kim et al., 2023). Taken together, these studies have either focused on developmental anatomy or on perturbations unrelated to defined locomotor regimes and have typically treated muscle structure as a static background to gait mechanics. To our knowledge, no previous work has systematically quantified region-specific myocyte geometry together with sarcomere organization in wild-type animals under distinct locomotor conditions such as sustained crawling versus swimming.

Crawling on agar imposes low-frequency, low-curvature bending on the body wall, whereas swimming in liquid requires higher-frequency undulations and greater mid-body curvature (Beron et al., 2015). Muscle-intrinsic mechanosensation has emerged as a key determinant of gait-appropriate force production (Kang et al., 2010; Wen et al., 2012; Thomas et al., 2024). Fazyl et al. (2025) showed that the mechanosensitive ion channel PEZO-1 localizes to the sarcolemma of body-wall myocytes, where it differentially modulates calcium dynamics and contractile output during swimming and crawling. Loss of PEZO-1 activity impairs swimming while modestly enhancing aspects of crawling, indicating that muscle mechanotransduction contributes to matching force generation to environmental load.

Beyond acute modulation of contractile activity, muscles remodel their internal architecture with chronic changes in use (Ferraro et al., 2014; Franchi et al., 2017; Pillon et al., 2020). In vertebrates,

School of Biological Sciences, Illinois State University, Normal, IL 61790, USA.

*Author for correspondence (avidal@ilstu.edu)

A.F., 0009-0007-5999-8265; A.G.V.-G., 0000-0001-5981-5528

Biology Open

sustained mechanical loading alters sarcomere number, spacing, and lattice geometry to optimize contractile efficiency through mechanotransduction pathways that sense strain and trigger adaptive responses (Højfeldt et al., 2023; Hoppeler, 2016). Chronic high-frequency, low-load activity such as endurance exercise promotes serial sarcomere addition and shifts fiber type composition toward oxidative metabolism, whereas high-force, low-frequency loading favors parallel addition of myofibrils and increased cross-sectional area (Pillon et al., 2020). These adaptations are mediated by mechanosensitive signaling cascades involving integrin-based adhesions, stretch-activated channels, and titin-based strain sensors that couple mechanical input to transcriptional programs regulating sarcomeric protein synthesis and degradation (Hoppeler, 2016). While acute exercise can modulate muscle gene expression and function in invertebrates (Laranjeiro et al., 2019), whether chronic developmental conditioning in distinct locomotor environments drives structural remodeling of myocyte geometry and sarcomere organization has remained largely unexplored. However, previous findings suggest that nematode muscles possess a dynamic, plastic contractile architecture capable of adapting to altered forces.

Muscles can adjust force, shortening velocity, and working range by changing cell geometry and sarcomere arrangement. In simple sliding filament terms, a longer myocyte packed with more sarcomeres in series can shorten faster over a larger excursion, while a wider myocyte with greater cross-sectional area tends to produce higher peak force. A myocyte with more sarcomeres in series can shorten faster than one of equal length containing fewer, longer sarcomeres, although this comes at the cost of peak force per unit cross-section (Hill, 1938). Sarcomere length sets the operating range of thin and thick filament overlap, and serial sarcomere number controls how many contractile units contribute to whole-cell shortening. Shape descriptors such as aspect ratio, anisotropy, and circularity translate these anatomical properties into quantitative indices: high anisotropy and low circularity indicate elongated, beam-like myocytes that are mechanically suited to large bending excursions, whereas more isotropic, broad cells resemble force-focused actuators. Within this framework, myocyte area reports the planar supply of contractile material, Feret length approximates the cell's working length, MinFeret captures its width and potential force-bearing capacity, and sarcomere density reports how tightly the contractile lattice is packed per unit area.

Here we focus on the healthy muscle response to the locomotor environment, isolating the effects of gait-related mechanical demand on myocyte morphology and sarcomeric organization. Based on the kinematics of swimming, which involves higher-frequency undulations and greater body curvature than crawling (Pierce-Shimomura et al., 2008; Beron et al., 2015), we hypothesized that sustained swimming would remodel the contractile lattice to produce adaptations favoring velocity and flexibility over pure force generation. Specifically, we predicted that swimming would induce: 1) global myocyte elongation (increased Feret diameter) to accommodate greater bending excursions, 2) selective reductions in cell width and cross-sectional area in regions experiencing highest curvature (the mid-body), 3) shorter average sarcomere length to shift toward faster contraction velocity, and 4) increased serial sarcomere number and density to maintain force production while optimizing for speed. These predictions follow from the principle that muscles composed of many short sarcomeres in series contract faster than muscles of equal length composed of fewer, longer sarcomeres (Hill, 1938), and from biomechanical analyses showing that undulatory swimming

imposes peak bending strain at the mid-body (Pierce-Shimomura et al., 2008). Using confocal imaging of phalloidin-stained myocytes from animals reared exclusively under crawling or swimming conditions, we quantified myocyte geometry, sarcomere length, and sarcomere number across anterior, mid-body, and posterior body regions. Our findings reveal coordinated, region-specific remodeling that supports the predicted functional adaptations. Integrated with prior behavioral (Vidal-Gadea et al., 2011; Pierce-Shimomura et al., 2008) and mechanotransduction (Fazyl et al., 2025) studies, these findings link neural gait selection and PEZO-1-mediated feedback to long-term structural plasticity of the contractile lattice, positioning *C. elegans* as a comprehensive model for dissecting how environmental mechanics shape muscle function from neural control through mechanotransduction to cellular architecture.

## RESULTS
### Experimental design and measurement strategy
To determine how sustained locomotor behavior influences muscle architecture, we compared body-wall myocytes from wild-type *C. elegans* reared exclusively under crawling or swimming conditions (Fig. 1A). In *C. elegans* this is made possible by the geometry of the body wall musculature. Specifically, body wall myocytes occur in four quadrants and consist of single cells containing a singular layer of non-overlapping sarcomeres. This allows the use of maximal projections from confocal stacks to capture the entire contractile machinery within each cell without the confound of multiple sarcomeres overlapping in the z plane (Gieseler et al., 2017). Confocal imaging of phalloidin-stained animals revealed clear differences in myocyte geometry and sarcomere organization between swimming and crawling conditions (Fig. 1D,E). Measurements were obtained from anterior, mid-body, and posterior myocytes (myocytes m1-10, m11-18, and m19-25, respectively). Each myocyte was outlined to determine geometric parameters (area, Feret, MinFeret, aspect ratio, circularity), and sarcomere number and length were quantified along the long axis (Fig. 2A). Derived metrics included sarcomere density (number per area) and anisotropy (Feret divided by MinFeret), which serves as a proxy for cell elongation. Our approach focused on myocyte area rather than volume. We decided to focus on this cellular dimension because it was the closest associated with the planar architecture of the contractile machinery.

### Myocyte geometry adapts to locomotor mode through region-specific thinning and global elongation
Swimming produced coordinated changes in myocyte geometry, driven by a combination of global adaptations and region-specific remodeling (Fig. 2A–F). Across regions, myocytes elongated substantially: the mean Feret diameter, a proxy for cell length, increased from 118.2±12.1 μm (mean±s.e.m.) in crawling head muscles to 134.6±8.5 μm in swimmers (*P*=0.0006). Similar elongation occurred in mid-body (from 127.1 μm to 136.3 μm, *P*=0.014) and posterior (from 117.0 μm to 129.8 μm, *P*=0.0006) regions (Fig. 2C). In contrast, cell width narrowed selectively in the mid-body. Minimum Feret, representing the myocyte's short axis, decreased from 16.0 μm in crawling worms to 12.8 μm in swimmers ($P<1\times10^{-6}$), whereas anterior and posterior widths were unchanged (Fig. 2D). Correspondingly, myocyte area in the mid-body region decreased by 154 μm² on average (1217±243 μm² versus 1063 ±207 μm²; *P*=0.032), whereas head and tail areas were unaffected (Fig. 2B). These shape adjustments yielded a consistent morphological signature. Anisotropy rose from 8.0 to 10.8 in the

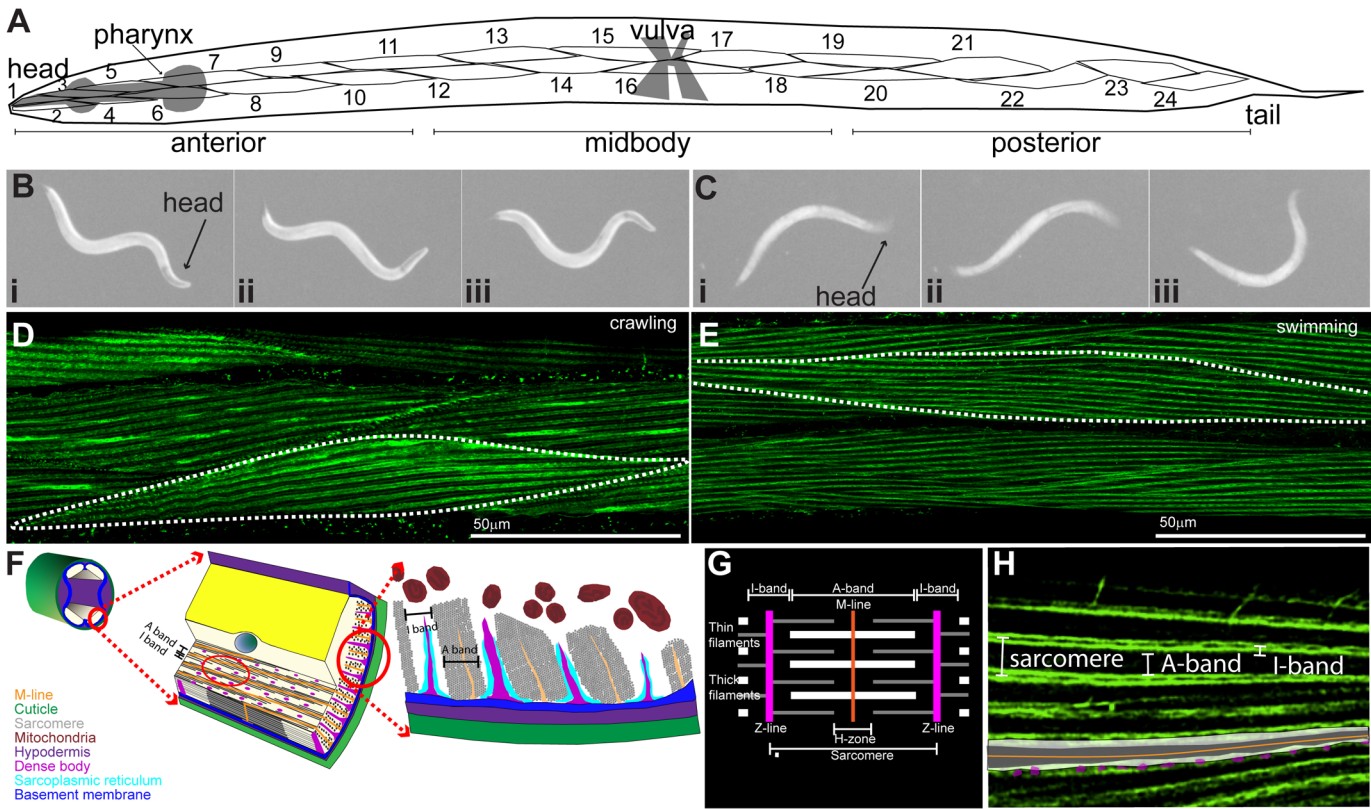

**Fig. 1. Experimental overview and measurement workflow.** (A) Schematic of *C. elegans* body-wall musculature showing anterior (head), mid-body, and posterior (tail) regions analyzed. (B) Representative bright-field stills of worms during crawling on agar (left) and (C) swimming in liquid (right), illustrating posture and curvature over time (i–iii). (D,E) Confocal micrographs of phalloidin-stained mid-body myocytes from crawling and swimming animals, respectively. Actin filaments delineate sarcomeres used for morphometry. (F) Schematic diagram of *C. elegans* body-wall musculature styled after Gieseler et al. (2017). (G) Typical sarcomere arrangement in mammalian striated muscle compared with analogous structures in *C. elegans* (H).

mid-body region ($P<10^{-6}$) and from 11.5 to 13.8 posteriorly ($P<0.0001$), reflecting stronger elongation (Fig. 2E). Circularity showed a significant locomotion effect ($P=0.080$ overall) modulated by region ($P=0.018$), decreasing notably in both mid and tail regions (from 0.223 to 0.178, $P<10^{-6}$; from 0.168 to 0.139, $P=0.0003$), confirming a shift toward more elongated and angular cell shapes (Fig. 2F). Higher circularity indicates more isotropic, force-focused cell morphology, while reduced circularity reflects beam-like cells better suited for repeated bending.

Together these data indicate that swimming promotes global myocyte elongation combined with selective mid-body thinning and shape reorganization, suggesting distinct mechanical loading patterns along the body axis.

### Sarcomere organization reflects coordinated activity-dependent remodeling

Swimming altered not only myocyte geometry but also the organization of the contractile lattice (Fig. 3B–D). Sarcomeres were significantly shorter in swimmers: average lengths decreased from 1.62 μm to 1.43 μm in the head ($P=0.003$), from 1.85 μm to 1.49 μm in the mid-body ($P<10^{-9}$), and from 1.57 μm to 1.36 μm in the tail ($P=7×10^{-5}$) (Fig. 3B). This widespread shortening indicates a global remodeling program rather than localized contraction artifacts.

Swimming also increased sarcomere number ($P=0.0006$), adding approximately 0.8 sarcomeres per myocyte in the head ($P=0.0013$) and 0.6 in the mid-body ($P=0.0022$), while the tail remained unchanged ($P=0.81$) (Fig. 3C). Sarcomere density showed a mild

overall increase ($P=0.42$) but a clear condition×region interaction ($P=0.021$), driven by a rise in mid-body region from 0.0066 μm$^{-2}$ to 0.0080 μm$^{-2}$ ($P=0.010$), suggesting serial addition and tighter packing (Fig. 3D).

Integrating the geometric and sarcomeric data reveals a coherent myocellular adaptation program. During swimming, myocytes globally elongate, and the contractile lattice simultaneously shifts toward more, shorter sarcomeres. This coordinated remodeling is most pronounced in the mid-body, where geometry (thinning and area reduction) and lattice organization (serial sarcomere addition) change together to optimize the myocyte for the high-frequency, high-flexibility contractions required for swimming, rather than the high-force output of crawling.

### Regional specialization and coordinated remodeling of cell size and sarcomere number

To test whether the geometric and sarcomeric adaptations documented above reflected coordinated changes within individual myocytes, we examined the relationship between myocyte area and sarcomere number. In anterior muscles, a modest positive correlation emerged when data were pooled across locomotor conditions ($r^2=0.33$, $P=0.005$, $n=22$), suggesting that larger anterior myocytes accommodate proportionally more sarcomeres in series (Fig. 4A). This relationship was not significant within crawling or swimming animals alone, indicating variability in how individual myocytes scale sarcomere number with size. In contrast, mid-body and posterior regions showed no significant correlation between area and sarcomere number (mid-body:

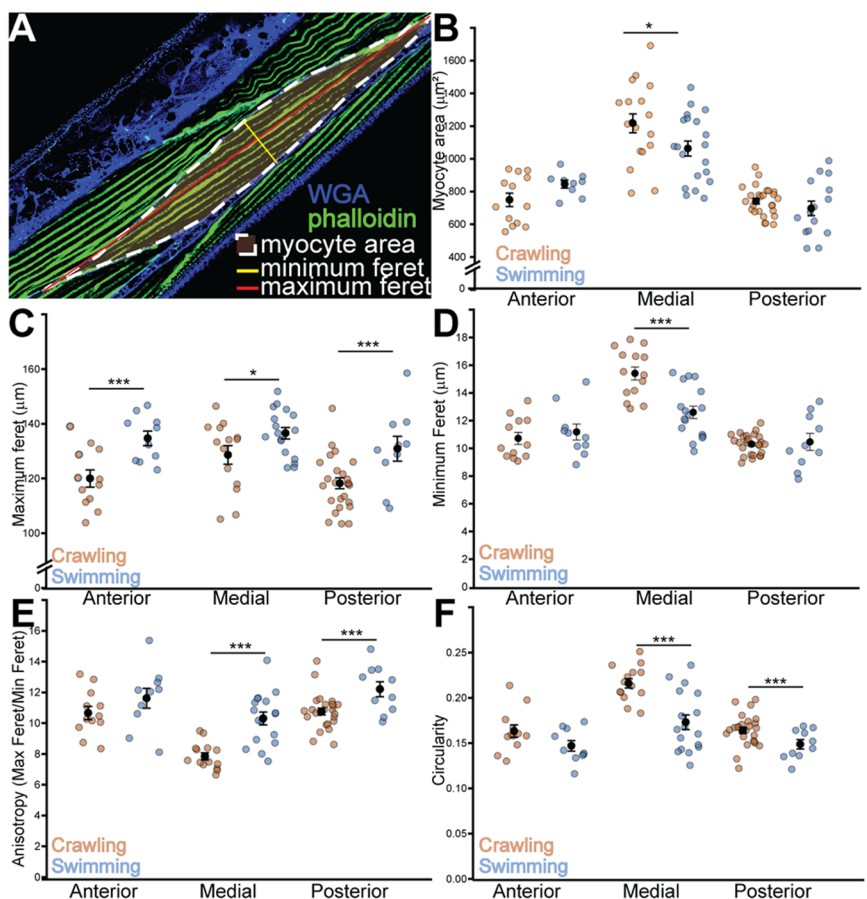

**Fig. 2. Myocyte geometry adapts to locomotor mode through global elongation and region-specific thinning.** (A) Representative confocal micrograph illustrating the quantitative analysis workflow. Body-wall muscle is stained with phalloidin (green) to visualize F-actin, and WGA (blue) stains cell boundaries. Overlays demonstrate the key geometric measurements taken for each myocyte: myocyte area (white dashed outline), maximum Feret diameter (red line), and minimum Feret diameter (yellow line). (B) Area. Region-dependent effect of swimming (LMM, significant condition×region interaction, $P$=0.033), with smaller mid-body areas in swimmers (simple effect, $P$=0.032). ($n$: Head C=13, S=9; Mid C=18, S=20; Tail C=26, S=15). (C) Feret (maximum diameter). Greater Feret in swimmers across regions (OLS-CR, main effect of condition, $P$=0.00019). ($n$: Head C=13, S=10; Mid C=18, S=18; Tail C=25, S=15). (D) MinFeret (minimum diameter). Selective mid-body thinning in swimmers (LMM, significant condition×region interaction, $P$=0.0027). ($n$: Head C=13, S=8; Mid C=18, S=20; Tail C=26, S=16). (E) Anisotropy (Feret/ MinFeret). Increased anisotropy in swimmers, most pronounced in mid-body region (OLS-CR, significant condition×region interaction, $P$=0.015). ($n$: Head C=13, S=9; Mid C=18, S=20; Tail C=22, S=14). (F) Circularity. Reduced circularity specifically in mid-body and tail myocytes of swimmers (OLS-CR, significant condition×region interaction; simple effects $P$<0.001 for Mid and Tail). ($n$: Head C=12, S=10; Mid C=16, S=20; Tail C=25, S=15). $N$=96-103 animals; $n$=number of myocytes. C=Crawl, S=Swim. Data are shown as mean±s.e.m. *$P$<0.05; **$P$<0.01; ***$P$<0.001.

r²=0.05, $P$=ns, $n$=31; posterior: r²=0.10, $P$=ns, $n$=36) (Fig. 4B,C), indicating that sarcomere addition in these regions occurs independently of cell size changes.

When all regions were pooled, both crawling (r²=0.57, $P$<0.0003, $n$=52) and swimming (r²=0.36, $P$<0.0007, $n$=37) animals displayed

significant positive correlations between area and sarcomere number (Fig. 4D). The strength of this pooled correlation reflects primarily the intrinsic differences between body regions rather than a tight within-region scaling relationship. Specifically, mid-body myocytes are naturally larger and contain more sarcomeres than

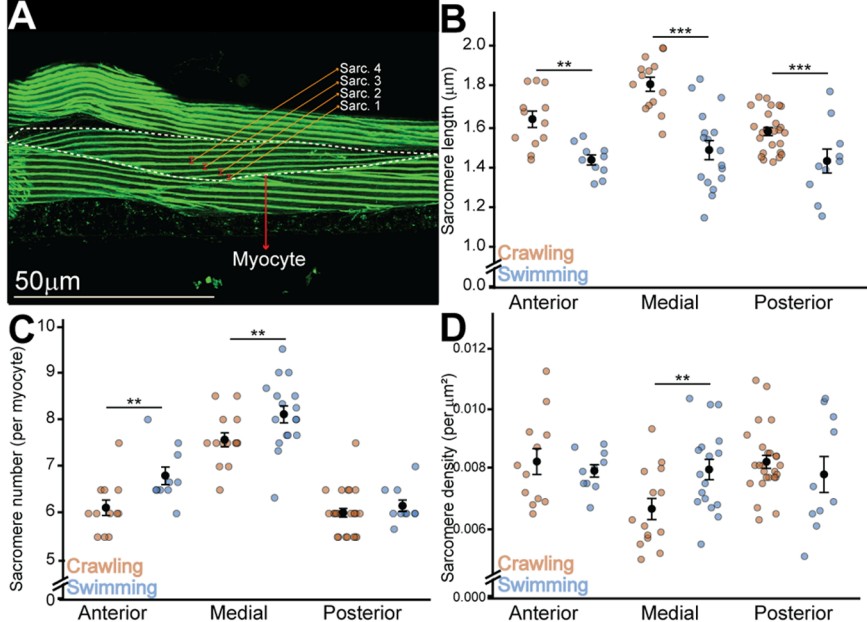

**Fig. 3. Sarcomere organization reflects coordinated activity-dependent remodeling.** (A) Representative confocal micrograph illustrating sarcomere measurement methodology. Phalloidin-stained body-wall muscle (green) shows the repeated Z-line pattern used to quantify sarcomere length and number. White dashed lines outline a single myocyte, and colored brackets indicate four consecutive sarcomeres (Sarc. 1-4) measured along the long axis of the cell. Scale bar: 50 μm. (B) Sarcomere length. Sarcomeres are shorter in swimmers across all regions (LMM, main effect of condition, $P$=0.003). (Ns: Head C=13, S=10; Mid C=18, S=20; Tail C=26, S=16). (C) Sarcomere number. Swimmers show higher sarcomere number in Head and Mid regions (OLS-CR, main effect of condition, $P$=0.0006), but this effect is absent in the tail (significant condition×region interaction for Tail, $P$=0.0019). (Ns: Head C=12, S=10; Mid C=15, S=20; Tail C=25, S=14). (D) Sarcomere density (per μm²). Swimming significantly increases sarcomere density in the mid-body region (LMM, significant condition×region interaction, $P$=0.021). (Ns: Head C=13, S=10; Mid C=18, S=20; Tail C=22, S=16). $N$=96-103 animals; $n$=number of myocytes. C=Crawl, S=Swim. Data are shown as mean±s.e.m. *$P$<0.05; **$P$<0.01; ***$P$<0.001.

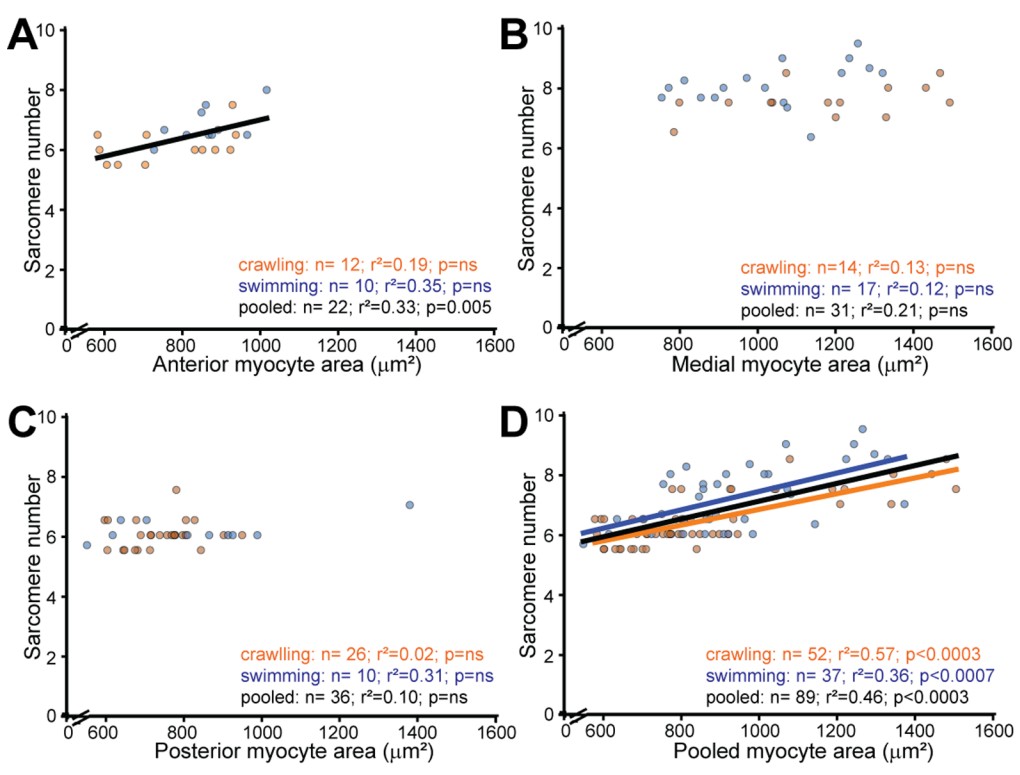

**Fig. 4. Myocyte area and sarcomere number show region-dependent scaling relationships.** Scatter plots show the relationship between sarcomere number (x-axis) and myocyte area (y-axis) in (A) anterior, (B) mid-body, and (C) posterior body regions, with (D) all regions pooled. Within individual regions, correlations are either weak (anterior: $r^2$=0.33, $P$=0.005) or absent (mid-body: $r^2$=0.05, $P$=ns; posterior: $r^2$=0.10, $P$=ns). When all regions are pooled, both crawling ($r^2$=0.57, $P$<0.0003) and swimming ($r^2$=0.36, $P$<0.0007) animals display significant positive correlations, reflecting primarily the intrinsic differences between body regions rather than tight within-region scaling. Data are individual myocytes after 1.5×IQR outlier removal ($N$=89 total: Anterior $n$=22, Mid-body $n$=31, Posterior $n$=36). Lines represent least-squares linear regression fits (pooled in black; crawlers in brown; swimmers in blue). The parallel condition-specific slopes in panel D indicate that swimming shifts sarcomere number and area through the regional remodeling documented in Figs 2 and 3 without fundamentally altering the scaling relationship.

anterior or posterior myocytes in both locomotor conditions, driving the overall positive association. The parallel slopes of the condition-specific regression lines indicate that swimming does not fundamentally alter how sarcomere number scales with area; instead, swimming shifts both parameters through the regional remodeling patterns documented in Figs 2 and 3. We tested non-linear regression models for these relationships but found that linear models provided adequate fits given our sample sizes.

Collectively, these data identify a distributed remodeling program. Swimming drives global myocyte elongation and sarcomere shortening across all body regions while simultaneously inducing dramatic region-specific changes in the mid-body, where muscles thin by approximately 3 µm in minimum diameter, lose approximately 150 µm² of cross-sectional area, and increase sarcomere density. This compartmentalized response, combining uniform adjustments for contraction speed with targeted mid-body remodeling, aligns with the mechanical demands of swimming, where curvature and bending strain peak at the mid-body (Pierce-Shimomura et al., 2008; Beron et al., 2015).

## DISCUSSION
Our results show that locomotor context shapes myocyte architecture through coordinated and region-specific remodeling. Across the body, swimming drove elongation of myocytes and robust sarcomere shortening, coupled with increased sarcomere number in anterior and mid-body regions. These adjustments are consistent with tuning the muscle lattice for high-frequency undulation and increased flexibility in liquid, as predicted by sliding filament theory (Hill, 1938).

The most striking adaptations localized to the mid-body, where muscles thinned, reduced area, lost circularity, and increased anisotropy. This structural narrowing coincided with a rise in sarcomere density and a marked shortening of individual

sarcomeres, forming a dense contractile lattice optimized for rapid curvature changes. Such compartmentalized remodeling aligns with mechanical analyses showing that curvature and bending strain peak in the mid-body region during swimming, demanding both flexibility and continuous force transmission (Pierce-Shimomura et al., 2008).

The correlation analysis between myocyte area and sarcomere number revealed that these two parameters scale together primarily at the level of regional identity rather than through tight within-myocyte coupling. Within individual body regions, the relationship was either weak (anterior) or absent (mid-body and posterior), yet when all regions were pooled, a strong positive correlation emerged. This pattern indicates that the remodeling program operates through region-specific set points rather than a simple rule linking cell size to sarcomere number. Mid-body myocytes, whether in swimmers or crawlers, are intrinsically larger and contain more sarcomeres than anterior or posterior myocytes, establishing distinct contractile configurations for each body segment. Swimming modulates these regional baselines, adding sarcomeres in anterior and mid-body regions while selectively thinning the mid-body, without fundamentally altering the regional scaling relationships. This compartmentalized strategy allows independent tuning of different mechanical properties (force output, shortening velocity, strain resistance) along the body axis, matching contractile architecture to segment-specific demands during undulatory locomotion.

These findings fit with a mechanotransduction framework in which activity and load history tune muscle structure. Prior work from our group and others points to stretch-activated channels, including PEZO-1, as candidate sensors of mechanical state (Komandur et al., 2023; Fazyl et al., 2025). The mid-body-dominant remodeling observed here provides a testable anatomical substrate for such sensing. In future experiments, manipulating PEZO-1 in muscle and measuring the same geometric and lattice

endpoints should reveal whether these channels are required to couple behavior to structural adaptation. More broadly, the pattern we document mirrors principles relevant to disease. In dystrophin deficiency, muscles face elevated mechanical stress and impaired force transmission (Petrof et al., 1993; Allen et al, 2016). An adaptation program that increases sarcomere number while thinning and reshaping cells in regions of highest strain may fail or misfire without proper membrane stabilization, offering a route to early dysfunction.

Our approach has limitations that set the scope of inference and point to future work. First, we measured myocyte area rather than volume. Our focus on area is based on the planar arrangement of the contractile machinery in *C. elegans* body wall myocytes. As such it closely matches the force production capacity of these cells. We worked from maximum intensity projections of z-stacks spanning the entire myocyte layer. Because body wall muscle in this animal presents a single myofibrillar layer under the membrane, this approach likely captured the dominant activity-induced changes, yet it cannot exclude depth-wise remodeling. Myocytes can adapt through different pathways and in addition to the myofibrillar (hypertrophic) growth targeted by our approach. In addition, they are able to undergo sarcoplasmic growth, which would result in an increase in cell volume without a corresponding increase in contractile machinery (Gezer et al., 2025). Such growth could affect other relevant myocyte components such as mitochondrial and sarcoplasmic reticulum volume which in turn can have important consequences for muscle function. For example, Kugelberg and Thornell found sarcoplasmic reticulum volume was correlated with contractile speed in vertebrate muscles (Kugelberg and Thornell, 1983). Future studies with volumetric reconstructions or light-sheet imaging could address this directly.

Second, we focused on day one adults. This decision aligns with common practice and facilitates comparison with prior literature. However, these animals continue to grow and remodel over subsequent days. Extending the analysis through day five adults would determine whether the patterns we report are stable features of the adult program or evolve with continued growth. Third, we labeled F-actin with phalloidin to quantify sarcomeres. This marker is well suited for lattice geometry but does not report complementary components that may remodel with activity, including myosin heavy chain, α-actinin, costameric proteins, and membrane complexes. Parallel labeling or genetic reporters will be important to resolve which molecular compartments track with the geometric and lattice changes. Fourth, the environmental contexts are not identical beyond mechanics. Animals raised and assayed in liquid and on agar engage distinct behavioral programs and may experience different oxygenation, feeding, and sensory inputs, as well as potential differences in metabolic energy expenditure between the two locomotor modes (Vidal-Gadea et al., 2011, 2012). We therefore cannot attribute every observed difference uniquely to mechanical challenge. What we can say is that worms develop and reproduce robustly in both environments, and the structural adjustments we measure are therefore physiologically relevant, even if multiple inputs contribute. Finally, one variable, circularity, required an ordinary least squares model with cluster-robust standard errors due to a near-singular random-effects structure after filtering. The direction and magnitude of the effect were consistent with the mixed-model results for the other variables, but this analytic caveat should be noted.

In sum, *C. elegans* body wall muscle exhibits a coordinated adaptation program that couples cell-level geometry to sarcomere-level organization, with the strongest remodeling in the mid-body region. This pattern provides a simple, quantitative framework for testing how specific mechanosensory pathways, including PEZO-1 (Komandur et al., 2023; Fazyl et al., 2025), and force-transmission scaffolds, including dystrophin-associated complexes, translate behavior into structural change.

## MATERIALS AND METHODS
### Strains and maintenance
Wild-type *Caenorhabditis elegans* (N2, Bristol strain) were used in all experiments. Animals were maintained on nematode growth medium (NGM) agar plates seeded with *Escherichia coli* OP50 at 20°C under standard conditions (Brenner, 1974). All assays used age-synchronized day-1 adults obtained by alkaline hypochlorite bleaching of gravid hermaphrodites followed by timed development on NGM plates.

### Locomotor regimens
To examine activity-dependent remodeling, animals were reared exclusively under one of two locomotor conditions from the L1 stage until the day of imaging. For crawling, worms developed on standard NGM agar plates seeded with OP50. For swimming, synchronized embryos were placed in 1.5 ml microcentrifuge tubes containing liquid NGM supplemented with OP50 (optical density approximately 0.8) and gently agitated on a nutator at 20°C to maintain oxygenation (Laranjeiro et al., 2019). Animals were transferred to fresh liquid media every 48 h.

### Sample preparation and fixation
Animals were washed three times in liquid NGM buffer, anesthetized with 10 mM sodium azide, and fixed in 4% paraformaldehyde in phosphate-buffered saline (PBS) for 15 min at 20°C. Fixed worms were transferred to poly-L-lysine-coated slides and permeabilized using the freeze-crack method (Duerr, 2013). Following permeabilization, samples were blocked for 30 min in 1% bovine serum albumin (BSA) in PBS and stained overnight at 4°C with iFluor 488-conjugated phalloidin (1:400; Cayman Chemical) to visualize filamentous actin. Slides were washed in PBS and mounted in ProLong Gold antifade reagent (Thermo Fisher Scientific).

### Confocal imaging
Fluorescent images were acquired on a Leica SP8 confocal microscope equipped with Lightning deconvolution (63×, NA 1.40 oil-immersion objective). For each animal, Z-stacks spanning entire body-wall myocytes were captured (0.17 µm step size, 15 to 30 slices). Maximum intensity projections were generated in Leica Application Suite X (LAS X, v3.5.5.19976) and exported as 16-bit TIFFs. Imaging was performed from anterior, mid-body, and posterior regions corresponding to muscle cells 1–10, 11–18, and 19–25, respectively (Gieseler et al., 2017). Imaging was performed on body-wall myocytes located in the left and right dorsal and ventral quadrants without distinction, as these quadrants are arranged symmetrically and prior work has not identified laterality-dependent differences in myocyte geometry or contractile properties in wild-type animals (Gieseler et al., 2017). For each animal, we typically analyzed three or four myocytes per body region (anterior, mid-body, posterior), ensuring that the majority of measurements represent independent biological replicates from different individual worms rather than technical replicates from the same animal.

### Image analysis
Quantitative analysis was performed using Fiji/ImageJ (Schindelin et al., 2012). For each myocyte, we measured parameters that report both cellular geometry and sarcomeric organization, selected to link morphology to predicted functional output. Cellular geometry measurements included: myocyte area (relates to force-producing capacity and total contractile material deployed in the imaging plane); perimeter (defines cell boundary complexity); Feret diameter (maximum cell dimension, approximates working length and maximal shortening excursion); MinFeret (minimum cell dimension, approximates cross-sectional width related to force-bearing capacity); major and minor axes from best-fit ellipse; aspect ratio (major/

minor axis); circularity ($4\pi\times$area/perimeter$^2$, where values near 1 indicate rounded cells and lower values indicate elongated, angular cells suited to bending); and solidity (ratio of cell area to convex hull area, reporting membrane complexity). The ratio of Feret to MinFeret, termed anisotropy, provides a simple index of cell elongation, with higher values indicating beam-like morphology. Sarcomeric measurements included: average sarcomere length, measured as the mean distance between consecutive Z-line actin peaks (identified from Plot Profile intensity measurements along the cell long axis; minimum five sarcomeres per cell); and sarcomere number, determined by counting Z-line intervals spanning the cell's long axis. All measurements were performed blind to locomotor condition. Derived variables included sarcomere density (sarcomere number divided by cell area), serial sarcomere density (sarcomere number divided by Feret length), and anisotropy (Feret divided by MinFeret). All measurements were performed blind to locomotor condition.

Derived variables were calculated to further characterize contractile lattice organization. Sarcomere density was computed as sarcomere number divided by myocyte area (sarcomeres/$\mu m^2$), reporting how tightly sarcomeres are packed within the available cellular footprint. Serial sarcomere density was calculated as sarcomere number divided by Feret diameter (sarcomeres/$\mu m$), indicating the linear packing density along the cell's long axis. Higher serial sarcomere density implies that a given cell length is subdivided into more, shorter sarcomeres, which is predicted to increase shortening velocity at the expense of peak force in the sliding filament model (Hill, 1938). All measurements were performed on maximum intensity projections of confocal Z-stacks, which is appropriate for *C. elegans* body-wall muscle because sarcomeres form a single non-overlapping layer beneath the sarcolemma (Gieseler et al., 2017), allowing two-dimensional projections to capture the full contractile apparatus without depth-related occlusion.

## Statistical analysis

All statistical analyses were conducted in R [version 4.5.0 (2025-04-11)] using a custom analysis pipeline. Data processing and visualization were performed using the tidyverse and ggplot2 packages. Prior to modeling, data were pre-processed using a robust, group-wise outlier removal procedure. This was performed independently for each dependent variable. Outliers were defined within each experimental group (for each unique condition×region combination, such as "Swim-Head"). This group-wise approach prevents data from one group (a high-variance group) from biasing outlier detection in another (a low-variance group). For a given variable, observations falling outside 1.5 times the interquartile range (IQR) of their specific group were excluded from that variable's analysis.

A robust, two-pronged hybrid modeling strategy was implemented to account for the hierarchical structure of the data (multiple measurements nested within each worm). This strategy ensured that the most statistically appropriate model was automatically selected for each dependent variable based on its specific data characteristics. The core model aimed to test the fixed effects of condition (Swim versus Crawl), region (Head, Mid, Tail), and their two-way interaction (condition*region).

We first attempted to fit a linear mixed model (LMM) using the lmerTest (Bates et al., 2015) package, with worm specified as a random intercept: Dependent Variable~condition*region+(1 | worm). This model was fit using restricted maximum likelihood (REML). The LMM was considered the appropriate model only if it met several criteria: the model converged without errors; the model was not 'singular', defined as the variance of the worm random effect being estimated at or near zero (<1e-8); and the pre-calculated intraclass correlation coefficient (ICC) for the worm group was greater than 0.01, indicating non-negligible clustering. For variables with significant, well-estimated inter-individual clustering, this LMM was retained.

If any of the LMM criteria failed, the model was considered inappropriate. In these cases, the analysis automatically fell back to a standard ordinary least squares (OLS) regression model: Dependent Variable~condition*region. This fallback was triggered for variables with no detectable inter-worm variance, such as Maximum Feret and Sarcomere number (ICC≈0).

*P*-values and confidence intervals were calculated using methods appropriate for the selected model. For LMMs, inference for the fixed effects was derived using the Kenward-Roger (KR) correction (ddf="Kenward-Roger"). The KR method is considered the gold standard for small-sample LMMs, as it provides a denominator degrees-of-freedom (DF) approximation and, critically, adjusts the fixed-effect standard errors to reduce small-sample bias. This was successfully applied to all LMMs, as seen in the final results (DF_method="Kenward-Roger"). For OLS Models, standard OLS *P*-values are invalid as they ignore the clustered data structure. Therefore, for all fallback OLS models, we calculated cluster-robust standard errors (using the sandwich and lmtest packages) by clustering on the worm identifier. This 'sandwich' estimator provides valid *P*-values and confidence intervals that are robust to the non-independence of observations from the same worm, even if an LMM could not be fit. The specific correction used was 'HC1' (DF_method="ClusterRobustHC1"). 'HC' stands for 'heteroskedasticity-consistent', and the '1' specifies a particular type of small-sample correction. This HC1 correction applies a DF adjustment to the standard error calculation, which prevents underestimation of the variance and provides more reliable *P*-values when the number of clusters (in this case, worms) is small. All statistical tests were two-tailed, and a *P*-value <0.05 was considered statistically significant.

Finally, to decompose significant interactions and directly test the effect of swimming versus crawling at each anatomical location, we conducted a post-hoc simple effects analysis using the emmeans package. We calculated the estimated marginal means (EMMs) for the 'Swim versus Crawl' contrast within each of the three regions (Head, Mid, Tail). This post-hoc analysis used a parallel modeling strategy. It fit the data to an LMM with Satterthwaite-approximated degrees of freedom, falling back to a standard OLS model if the LMM was invalid. This ensured the *P*-values for these specific contrasts were also robustly estimated and properly accounted for the data's structure.

Across all experiments, we analyzed a total of 97 individual animals (52 crawling, 45 swimming), yielding 215 myocytes before outlier removal (112 from crawling animals, 103 from swimming animals). During our analysis we measured between two and four myocytes per body region and between one and two body regions per animal (see Supplementary dataset 1). The outlier removal step removed on average less than 5% of the dataset (1.9–6.7%) for any one metric. Please refer to Supplementary dataset 2 for exact sample sizes for each variable and body region. Sample sizes are also reported in figure legends. Raw data is provided in Supplementary dataset 1. Full statistical analysis and the outlier log are provided in Supplementary dataset 2.

## Use of artificial intelligence (AI) tools

Artificial intelligence tools were used (Claude) to check grammar and writing clarity, and to verify adherence to journal formatting standards. No AI tools were used for data collection, analysis, interpretation, or generation of scientific content. All scientific conclusions and interpretations are solely those of the authors.

## Data availability

All raw measurements and statistical outputs are provided as a supplement (supplementary information) and figure source data. Representative imaging stacks and analysis scripts are available at https://figshare.com/s/8ea23d743d7c3e5740d0.

## Acknowledgements

Some strains were provided by the CGC, which is funded by NIH Office of Research Infrastructure Programs (P40 OD010440).

## Competing interests

The authors declare no competing or financial interests.

## Author contributions

Conceptualization: A.G.V.-G.; Data curation: A.F., S.K., A.G.V.-G.; Formal analysis: A.F., A.A., S.K., A.G.V.-G.; Funding acquisition: A.G.V.-G.; Investigation: A.F., S.K.; Methodology: A.F., A.G.V.-G.; Project administration: A.G.V.-G.; Resources: A.G.V.-G.; Supervision: A.G.V.-G.; Validation: A.F.; Visualization: A.G.V.-G.; Writing – original draft: A.F., A.G.V.-G.

## Funding

This work was supported by NIH (National Institute of Arthritis and Musculoskeletal and Skin Diseases) grant 2R15AR068583-02 to A.G.V.-G. Open Access funding provided by Illinois State University. Deposited in PMC for immediate release.

Biology Open

### Data and resource availability

All raw measurements, statistical outputs, representative imaging stacks, and analysis scripts are available at https://figshare.com/s/8ea23d743d7c3e5740d0. All relevant data and details of resources can be found within the article and its supplementary information.

### Peer review history

The peer review history is available online at https://journals.biologists.com/bio/lookup/doi/10.1242/bio.062371.reviewer-comments.pdf

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
