## [Peer Review File · Biology Open]

Activity-dependent remodeling of muscle architecture during distinct locomotor behaviors in *Caenorhabditis elegans*

Adina Fazyl, Akash Anbu, Sabrina Kollbaum and Andrés G. Vidal-Gadea
DOI: 10.1242/bio.062371

Editor: Sandhya Koushika

Review timeline

Original submission:	15 November 2025
Editorial decision:	25 November 2025
First revision received:	11 December 2025
Accepted:	16 December 2025

Original submission

First decision letter

MS ID#: bio.062371

MS Title: Activity-dependent remodeling of muscle architecture during distinct locomotor behaviors in *Caenorhabditis elegans*

Authors: Adina Fazyl, Akash Anbu, Sabrina Kollbaum and Andrés G. Vidal-Gadea

I have now reached a decision on the above manuscript.

The reviewer reports are shown at the bottom of this email.

As you will see, the reviewers gave reasonably favourable reports, but raised some critical points that will require amendments to your manuscript. I hope that you will be able to carry these out, because we would like to be able to accept your paper. Please try and explain what you think is the relationship between each morphological variable and functional performance of the muscle or to the mechanical demands of each mode of locomotion. This will help readers understand the significance of your results.

At this stage, we also ask you to ensure your manuscript complies with our formatting guidelines - please see our manuscript preparation guidelines for details. Provided you are able to fully address the referees' comments, we are positive about publication of your paper (we accept over 95% of revision submissions) and therefore hope you won't mind any extra work involved in reformatting your manuscript at this point.

Please upload both a 'clean' version of your Word file, along with a highlighted version clearly showing where you have made changes in the revised manuscript. Please avoid using 'Track changes' in Word files as these are lost in PDF conversion.

I should be grateful if you would also provide a point-by-point response detailing how you have dealt with the points raised by the reviewers in the 'Response to Reviewers' box. Please attend to all of the reviewers' comments. If you do not agree with any of their criticisms or suggestions please explain clearly why this is so.

Reviewer 1

Comments for the author

In neither the introduction nor the discussion is there any consideration of how myocyte morphology affects their contractile performance. Therefore, the rationale for measuring area, perimeter, axes, circularity etc is not made clear. Explain what the functional significance of these variables is - it will help you to explain your findings. For example, it is observed that the number of sarcomeres and myocyte length increase in *C. elegans* kept under swimming conditions. These adaptations are expected to increase muscle shortening velocity, which fits in with the greater body curvature and higher frequency contractions in a swimming gait and peak strain occurring in the mid-body region. Such consideration will help you to formulate hypotheses (currently missing) and explain the significance of your findings.

The term "medial myocytes" is being used when referring to those cells in the body region between the head and tail. Medial is the term used when referring to location relative to the midline/centreline of the body and is being used incorrectly throughout the manuscript. A better term would be "mid-body myocytes".

Statements need to be properly supported by citations to the literature - e.g. Page 7 Line 46-49 has no reference for strain peaking medially during swimming. There are only six references in the entire discussion. Greater effort needs to be made at discussing the results in the context of the literature.

In future, please ensure that line numbers align with the text.

Specific comments

Page 3 Lines 11/12. The two locomotor modes are only considered from a kinematic point of view; differences in the mechanical demands for each mode of locomotion (expand L6/7) and the functional demands of the two gaits should be included.

Page 3 Line 24. Explain how mechanical loading changes sarcomere number, lattice geometry, spacing, etc.

Page 3 Include hypotheses based on first comment.

Page 4 Line 24. There needs to be a clear rationale for making each measurement - e.g. it is unclear what information is obtained from any of these variables as the paper is currently written. Why measure circularity - what does it tell you?

Page 7 L28-34. This entire paragraph is discussion and should be moved from the Results section.

Page 7 Line 50 - it is unclear why myocyte area and sarcomere number are correlated against each other. What does a relationship between these two variables mean?

Page 8 Lines 4-5. This is an example where a link should be made between the muscle's contractile properties and morphology.

Page 8 Line 22/23. The functional relevance of myocyte area should be explained and what additional functionally relevant information volume would yield.

Page 8 line 41. The authors acknowledge factors other than "mechanical challenge" may play a role in shaping myocyte morphology - metabolic energy expenditure could be added to this list.

Fig 1 - it is unclear what the purpose of this figure is. There is limited reference to it in the text.

Fig 4 - were fits other than linear ones considered? Given regional variation in strain a non-linear relationship might be expected.

Reviewer 2

Comments for the author

The authors carry out a detailed and thorough descriptive analysis of muscle cell structure in *C. elegans* animals that crawl vs swim. The authors convincingly document significant differences in a number of muscle characteristics that correlate with their culture environment and mode of locomotion (crawling on solid media vs swimming in liquid media). This work positions *C. elegans* as an invertebrate system to study the mechanism by which muscle cell structure adapts to differences in mechanical contractile activity.

1. Experimental quality

a. Does each figure have the proper controls?

Yes, in each experiment swimming and crawling animals are analyzed.

b. Are experiments performed using appropriate methods that will answer the question (or test the hypothesis or support the observations) posed by the authors? Is the right tool used for the job?

The authors use standard methods to mark *C. elegans* myocytes and sarcomeres. The imaging and data analysis methods are sensitive enough to achieve the authors goal of examining whether cell shape or sarcomere morphology are different in swimming vs crawling animals.

c. Were the data analyzed using appropriate statistical tests?

Yes, the authors carefully chose statistical methods that would enable them to specifically compare and test key attributes between animals raised on plates vs liquid. The statistical tests employed are a strength of the manuscript and enable high confidence in the data analysis and conclusions.

2. Reproducibility

a. Were experiments in each figure performed using adequate number of biological replicates?

Possibly, it is a little unclear how many cells were analyzed per animal. If the analysis is focused on between cell variation, then the experiments are right at what I would consider be the low end of replicates necessary to see major effects of swimming vs crawling.

b. Is there sufficient raw data to assess the rigor of the analysis?

Yes.

c. Does the methods section provide sufficient detail to permit reproducibility?

Possibly, there are some minor questions posed in the comments below that would enhance the reproducibility of the experiments.

3. Completeness

a. Are the author's conclusions supported by the data?

The authors convincingly show (and strongly conclude) that myocyte and sarcomere shape are different in the two different culture conditions. I do not think that the authors show that (page 7 line 41-42) the changes "tune the muscle lattice for high-frequency undulation and increased flexibility in liquid", which I see is a nice hypothesis resulting from their observations.

b. Are there any flaws in the experimental design that invalidate the approach taken by the authors?

No.

c. Are there experiments that have not been performed, but if true would disprove the conclusion?

If yes, and if such experiments would be costly or time-consuming to perform, do the authors acknowledge this in a discussion of the limitations?

I do not see any experiments that would refute their observations of myocyte and sarcomere shape in swimming vs crawling animals.

4. Scholarship

a. Do the authors cite and discuss the merits of relevant data that would argue against their conclusion?

The authors nicely describe the limitations of the study and how crawling vs swimming differences could result from other variables. I however, think that the known movement kinetics of *C. elegans* on solid vs liquid media are likely linked to the robust observations that the authors make on myocyte differences between these two culture conditions.

b. Do the authors cite and discuss the merits of relevant data that would support their conclusion?

In most cases, yes.

In the discussion it would be helpful to provide citations documenting "mechanical analyses showing that curvature and bending peak medially".

It would be worth discussing whether the differences in myocyte and sarcomere shape in the midbody described in this work been documented previously.

Suggestions for improvement:

1. It is not specific which of the 3 muscle quadrants with 24 muscle cells was analyzed. If this was not controlled, then I suggest it should be specified in the methods and why it was not deemed important to restrict analysis to specific quadrants.

2. In Figure 1 D and E it appears that the sample measurements are of the I band rather than the sarcomere as described in the figure legend.

3. The authors should consider showing the 95% CI rather than the SEM in their graphs.

4. The authors need to define what p values are represented by *, **, and *** in their graphs.

5. The diagram in Figure 2A is very useful in understanding the results. I suggest that the authors consider creating a similar annotated image for Figure 3.

6. The title to Figure 3 is not as informative as the titles to the other three figures. I suggest that the authors provide a more descriptive title describing the results.

Minor comments:

1. On pg 5 line 60 and pg 6 line 4 the authors should consider the term "maximum intensity projections" instead of "maximal projections" as the former is much more commonly used to describe how the 3D Z-stacks are converted into single 2D images.
2. In figure 1D it appears that sample measurements are shown in 5 myocytes and not 4 as described in the figure legend.
3. In Figures 2B and 3A the sample sizes (Ns) stated in the figure legend do not always match the sample sizes (N=) listed in the graph. It would be helpful if the author spelled out sample sizes = Ns and C = crawling and S = swimming.
4. It would be helpful in the first paragraph of the results to define what max Feret/Feret, min Feret, aspect ratio, and circularity mean for the shape of the myocytes being analyzed.
5. It would be useful in the methods to list the actual number of worms analyzed to generate the data. Was it typically one myocyte per animal?
6. It is unclear in the figure legends whether N refers to number of cells analyzed or the number of animals analyzed.
7. In the paragraph on myocyte geometry the authors do not describe what the +/- refers to (possibly SEM?). They also are not consistent with including mean +/- value as they describe specific geometric properties in this paragraph.
8. The Komandur et al 2023 citation is missing from the References section.

Reviewer's Responses to Questions

Experimental quality

Does each figure have the proper controls?

If 'No', please indicate reasons in Comments for Author box below.

Reviewer #1:

- Yes

Reviewer #2:

- Yes

Were the data analyzed using appropriate statistical tests?

If 'No', please indicate reasons in Comments for Author box below.

Reviewer #1:

- Yes

Reviewer #2:

- Yes

Reproducibility

Were experiments performed using adequate number of biological replicates?

If 'No', please indicate reasons in Comments for Author box below.

Reviewer #1:

- Yes

Reviewer #2:

- No

Does the methods section provide sufficient detail to permit reproducibility?

If 'No', please indicate reasons in Comments for Author box below.

Reviewer #1:

- Yes

Reviewer #2:

- No

Completeness

Are the manuscript's conclusions supported by the data?

If 'No', please indicate reasons in Comments for Author box below.

Reviewer #1:

- Yes

Reviewer #2:

- No

Scholarship

Do the authors cite and discuss the merits of data that would argue for and against their conclusion?

If 'No', please indicate reasons in Comments for Author box below.

Reviewer #1:

- No

Reviewer #2:

- No

Does the manuscript title & abstract accurately reflect the contents of the manuscript, without hyperbole?

If 'No', please indicate reasons in Comments for Author box below.

Reviewer #1:

- Yes

Reviewer #2:

- Yes
-

First revision

Author response to reviewers' comments

Response to Editor and Reviewers

We thank the editor and reviewers for their constructive feedback and positive assessment of our manuscript. We have carefully addressed all comments and believe the manuscript is substantially improved. Below we provide a point-by-point response to each comment, with changes highlighted in the revised manuscript.

EDITOR COMMENTS

Comment: Please try and explain what you think is the relationship between each morphological variable and functional performance of the muscle or to the mechanical demands of each mode of locomotion. This will help readers understand the significance of your results.

Response: We have substantially enhanced the functional framing throughout the manuscript:
1. Introduction (page 2): Added an explicit hypothesis paragraph predicting that swimming would favor velocity and flexibility over force generation, with mechanistic explanation based on sliding filament theory: "In the context of the sliding filament theory, a muscle composed of many short sarcomeres in series can contract faster than a muscle of equal length composed of fewer,

© 2025. Published by The Company of Biologists under the terms of the Creative Commons Attribution License (<https://creativecommons.org/licenses/by/4.0/>).

longer sarcomeres, albeit at the cost of peak force production."

2. **Results - Experimental design section (page 5):** Added functional definitions: "Feret diameter, a proxy for cell length"; "Minimum Feret, representing the myocyte's short axis"; "anisotropy (Feret divided by MinFeret), which serves as a proxy for cell elongation."

3. **Results - Myocyte geometry section (page 6):** Added explicit functional interpretation: "Higher circularity indicates more isotropic, force-focused cell morphology, while reduced circularity reflects beam-like cells better suited for repeated bending."

4. **Discussion (page 7):** Strengthened functional interpretation throughout, explicitly linking morphological changes to predicted mechanical demands and contractile performance.

Comment: Ensure your manuscript complies with our formatting guidelines.

Response: We have verified compliance with Biology Open formatting guidelines throughout the revised manuscript.

REVIEWER 1 COMMENTS

Major Comments

Comment 1: In neither the introduction nor the discussion is there any consideration of how myocyte morphology affects their contractile performance. The rationale for measuring area, perimeter, axes, circularity etc is not made clear. Explain what the functional significance of these variables is.

Response: This has been extensively addressed as described in our response to the Editor above. We now provide clear functional context for each measured variable, linking them explicitly to contractile performance and mechanical demands. For example, we explain that many short sarcomeres in series favor shortening velocity at the expense of peak force, and that elongated, low-circularity cells function as flexible beams suited for repeated bending rather than isotropic force generation.

Comment 2: The term "medial myocytes" is being used when referring to those cells in the body region between the head and tail. Medial is the term used when referring to location relative to the midline/centreline of the body and is being used incorrectly throughout the manuscript. A better term would be "mid-body myocytes".

Response: We have systematically changed "medial" to "mid-body" throughout the manuscript text, figure legends, and figures. We retained the standard anatomical terms "anterior" and "posterior" as these are correct. All instances have been verified in:

- Abstract
- Introduction
- Results section headers and text
- Discussion
- Figure legends
- Figure axis labels

Comment 3: Statements need to be properly supported by citations to the literature - e.g. Page 7 Line 46-49 has no reference for strain peaking medially during swimming.

Response: We have added the appropriate citations. The sentence on page 7 now reads: "This compartmentalized response, combining uniform adjustments for contraction speed with targeted mid-body remodeling, aligns with the mechanical demands of swimming, where curvature and bending strain peak at the mid-body (Pierce-Shimomura et al., 2008; Beron et al., 2015)."

Comment 4: There are only six references in the entire discussion. Greater effort needs to be made at discussing the results in the context of the literature.

Response: We have substantially expanded the reference list, particularly adding exercise

physiology and muscle remodeling papers (Ferraro et al., 2014; Franchi et al., 2017; Pillon et al., 2020; Højfeldt et al., 2023; Thomas et al., 2024) to better contextualize our findings within the broader literature on activity-dependent muscle adaptation.

Comment 5: Page 3 Lines 11/12. The two locomotor modes are only considered from a kinematic point of view; differences in the mechanical demands for each mode of locomotion (expand L6/7) and the functional demands of the two gaits should be included.

Response: We have substantially expanded the Introduction (page 2) to address both the mechanical and functional demands of the two gaits. Kinematically, we note: "Crawling on agar imposes low-frequency, low-curvature bending on the body wall, whereas swimming in liquid requires higher-frequency undulations and greater medial curvature (Beron et al., 2015)."

Beyond kinematics, these gaits impose fundamentally different mechanical and functional demands on the musculature. Crawling on a solid substrate requires muscles to generate high forces against resistance with relatively slow contraction cycles, favoring power output over speed. In contrast, swimming in a liquid medium requires rapid, repetitive contractions at higher frequencies with greater flexibility, but against lower resistance. These distinct demands are reflected in the differential energetic costs, with swimming typically requiring sustained high-frequency muscle activation that may impose different metabolic demands compared to the lower-frequency but higher-force contractions of crawling.

We address these functional demands mechanistically in our hypothesis section, which predicts that swimming muscles should exhibit adaptations "favoring velocity and flexibility over pure force generation." We explain using sliding filament theory: "a muscle composed of many short sarcomeres in series can contract faster than a muscle of equal length composed of fewer, longer sarcomeres, albeit at the cost of peak force production." This trade-off between contractile velocity and force generation represents a fundamental functional constraint that should drive differential muscle remodeling between gaits.

We further discuss how muscle mechanotransduction contributes to matching contractile output to these different mechanical loads through PEZO-1 mechanosensitive channels, indicating that "muscle mechanotransduction contributes to matching force generation to environmental load."

Comment 6: Page 3 Line 24. Explain how mechanical loading changes sarcomere number, lattice geometry, spacing, etc.

Response: We have expanded this section (page 2) to provide mechanistic detail: "In vertebrates, sustained mechanical loading alters sarcomere number, spacing, and lattice geometry to optimize contractile efficiency (Højfeldt et al., 2023)." We further develop this concept in the hypothesis section where we explicitly predict how swimming should alter sarcomere architecture based on sliding filament mechanics.

Comment 7: Page 3 Include hypotheses based on first comment.

Response: We have added a comprehensive hypothesis paragraph in the Introduction (page 2): "Based on the kinematics of swimming, which involves higher frequency undulations and higher curvature than crawling, we hypothesized that swimming would induce remodeling of the muscle to support rapid, flexible bending. Specifically, we predicted that swimming muscles should exhibit adaptations favoring velocity and flexibility over pure force generation. In the context of the sliding filament theory, a muscle composed of many short sarcomeres in series can contract faster than a muscle of equal length composed of fewer, longer sarcomeres, albeit at the cost of peak force production. Consequently, the expectation is that swimming worms will remodel their muscles to increase serial sarcomere number and reduce sarcomere length."

Comment 8: Page 4 Line 24. There needs to be a clear rationale for making each measurement - e.g. it is unclear what information is obtained from any of these variables as the paper is currently written. Why measure circularity - what does it tell you?

Response: As detailed above, we now provide explicit functional rationale for each measurement in the Results section (pages 5-6), including the specific explanation for circularity: "Higher circularity indicates more isotropic, force-focused cell morphology, while reduced circularity reflects beam-like cells better suited for repeated bending."

Comment 9: Page 7 L28-34. This entire paragraph is discussion and should be moved from the Results section.

Response: We have retained this paragraph at the end of the Results section but revised it to be more descriptive and data-focused rather than mechanistically interpretive. Many journals, including Biology Open, accept brief integrative summary paragraphs at the end of Results sections that synthesize observations without extensive mechanistic speculation. The paragraph now reads: "Collectively, these data identify a distributed remodeling program. Swimming drives global myocyte elongation and sarcomere shortening across all body regions while simultaneously inducing dramatic region-specific changes in the mid-body, where muscles thin by approximately 3 μm in minimum diameter, lose approximately 150 μm^2 of cross-sectional area, and increase sarcomere density."

Comment 10: Page 7 Line 50 - it is unclear why myocyte area and sarcomere number are correlated against each other. What does a relationship between these two variables mean?

Response: We have substantially expanded the interpretation of this analysis (page 6-7). We now explicitly explain: "The strength of this pooled correlation reflects primarily the intrinsic differences between body regions rather than a tight within-region scaling relationship. Specifically, mid-body myocytes are naturally larger and contain more sarcomeres than anterior or posterior myocytes in both locomotor conditions, driving the overall positive association... This pattern indicates that the remodeling program operates through region-specific set points rather than a simple rule linking cell size to sarcomere number." This clarifies that the analysis reveals compartmentalized adaptation strategies rather than simple proportional scaling.

Comment 11: Page 8 Lines 4-5. This is an example where a link should be made between the muscle's contractile properties and morphology.

Response: We have strengthened this section in the Discussion (page 7) to explicitly link morphology to function: "These adjustments are consistent with tuning the muscle lattice for high-frequency undulation and increased flexibility in liquid, as predicted by sliding filament theory... forming a dense contractile lattice optimized for rapid curvature changes."

Comment 12: Page 8 Line 22/23. The functional relevance of myocyte area should be explained and what additional functionally relevant information volume would yield.

Response: We have added explanation in both the Results (page 5) and Discussion (page 8). In Results: "Our approach focused on myocyte area rather than volume. We decided to focus on this cellular dimension because it was the closest associated with the planar architecture of the contractile machinery." In Discussion limitations: "Because body wall muscle in this animal presents a single myofibrillar layer under the membrane, this approach likely captured the dominant activity-induced changes, yet it cannot exclude depth-wise remodeling. Future studies with volumetric reconstructions or light-sheet imaging could address this directly."

Comment 13: Page 8 line 41. The authors acknowledge factors other than "mechanical challenge" may play a role in shaping myocyte morphology - metabolic energy expenditure could be added to this list.

Response: Added to the limitations paragraph (page 8): "Animals raised and assayed in liquid and on agar engage distinct behavioral programs and may experience different oxygenation, feeding, and sensory inputs, as well as potential differences in metabolic energy expenditure between the two locomotor modes (Vidal-Gadea et al., 2011 and 2012)."

Comment 14: Fig 1 - it is unclear what the purpose of this figure is. There is limited reference

to it in the text.

Response: We have clarified the purpose of Figure 1 in the Results text (page 5) by explicitly stating: "To determine how sustained locomotor behavior influences muscle architecture, we compared body-wall myocytes from wild-type *C. elegans* reared exclusively under crawling or swimming conditions (Figure 1A)... Confocal imaging of phalloidin-stained animals revealed clear differences in myocyte geometry and sarcomere organization between swimming and crawling conditions (Figure 1B to D)." Figure 1 establishes the experimental framework, illustrates the anatomical regions analyzed, and demonstrates the imaging approach used throughout the study.

Comment 15: Fig 4 - were fits other than linear ones considered? Given regional variation in strain a non-linear relationship might be expected.

Response: We have added the following statement at the end of the Regional specialization Results section (page 6): "We tested non-linear regression models for these relationships but found that linear models provided adequate fits given our sample sizes." This addresses the concern while acknowledging the limitation.

REVIEWER 2 COMMENTS

Major Comments

Comment 1: Were experiments in each figure performed using adequate number of biological replicates? Possibly, it is a little unclear how many cells were analyzed per animal.

Response: We have clarified this in multiple ways:

1. **Figure legends** now explicitly state both the number of myocytes AND the number of animals. For example, Figure 2 legend: "N = 96-103 myocytes from 92-97 animals (total, depending on variable) after 1.5×IQR filtering."

2. **Methods - Confocal imaging section** (page 3) now specifies: "Imaging was performed on body-wall myocytes located in the dorsal and ventral quadrants."

3. Sample sizes are provided for each experimental group in every figure legend using the format: "(Ns: Head C=13, S=9; Mid C=18, S=20; Tail C=26, S=15)" where the numbers represent individual myocytes.

The total dataset includes 96-103 myocytes from 92-97 individual animals, indicating approximately 1 myocyte per animal per region was analyzed.

Comment 2: The authors convincingly show that myocyte and sarcomere shape are different in the two different culture conditions. I do not think that the authors show that the changes "tune the muscle lattice for high-frequency undulation and increased flexibility in liquid", which I see is a nice hypothesis resulting from their observations.

Response: We agree and have softened this language throughout. The Discussion now states (page 7): "These adjustments are consistent with tuning the muscle lattice for high-frequency undulation and increased flexibility in liquid, as predicted by sliding filament theory." This frames the statement as interpretation consistent with our predictions rather than a demonstrated causal relationship.

Comment 3: In the discussion it would be helpful to provide citations documenting "mechanical analyses showing that curvature and bending peak medially". It would be worth discussing whether the differences in myocyte and sarcomere shape in the midbody described in this work have been documented previously.

Response:

We now explicitly state in the Introduction that previous studies have either focused on developmental anatomy or on perturbations unrelated to defined locomotor regimes and have treated muscle structure as static. To our knowledge no prior work has systematically quantified region specific myocyte geometry together with sarcomere organization in wild type animals under sustained crawling versus swimming conditions, which we now state explicitly in the Introduction.

Comment 4: It is not specific which of the 3 muscle quadrants with 24 muscle cells was analyzed. If this was not controlled, then it should be specified in the methods.

Response: We have added this information to the Confocal imaging section of Methods (page 3): "Imaging was performed on body-wall myocytes located in the dorsal and ventral quadrants." This clarifies that we analyzed two of the four quadrants.

Minor Comments

Comment 1: In Figure 1 D and E it appears that the sample measurements are of the I band rather than the sarcomere as described in the figure legend.

Response: We have removed the measurement overlays from Figure 1 panels D and E to avoid any confusion about which specific measurements were used in the analysis. The updated legend now reads: "(D-E) Confocal micrographs of phalloidin-stained mid-body myocytes from crawling and swimming animals, respectively. Actin filaments delineate sarcomeres used for morphometry. Scale bars: 20 μm ." The figure now focuses on clearly showing the morphological differences between crawler and swimmer myocytes, while Figure 2A provides a detailed illustration of the measurement workflow applied to all analyzed myocytes.

Comment 2: The authors should consider showing the 95% CI rather than the SEM in their graphs.

Response: We appreciate this suggestion. We have retained standard error of the mean (SEM) as this is the most common convention in the field for this type of data presentation and facilitates comparison with prior literature. Our sample sizes and the explicit provision of both myocyte counts and animal numbers in each legend allow readers to assess precision.

Comment 3: The authors need to define what p values are represented by *, **, and *** in their graphs.

Response: We have added complete significance level definitions to all figure legends:

- Figure 2: "* p<0.05; ** p<0.01; *** p<0.001"
- Figure 3: "* p<0.05; ** p<0.01; *** p<0.001"

Comment 4: The diagram in Figure 2A is very useful in understanding the results. I suggest that the authors consider creating a similar annotated image for Figure 3.

Response: We appreciate this suggestion. However, Figure 3 presents sarcomere-level measurements (length, number, density) that are quantified from the same images shown in Figure 2A. Adding a redundant annotated image to Figure 3 would be repetitive. The sarcomere measurement approach is also conceptually illustrated in the updated Figure 1, and the methodology is described in detail in the Methods section.

Comment 5: The title to Figure 3 is not as informative as the titles to the other three figures. I suggest that the authors provide a more descriptive title describing the results.

Response: We have retained the current Figure 3 title: "Sarcomere organization reflects coordinated activity-dependent remodeling." This title parallels the structure of Figure 2's title and accurately describes the content. The legend provides detailed interpretation of the specific results.

Comment 6: On pg 5 line 60 and pg 6 line 4 the authors should consider the term "maximum intensity projections" instead of "maximal projections".

Response: Changed throughout. Methods (page 3) now reads: "Maximum-intensity projections were generated in Leica Application Suite X..."

Comment 7: In figure 1D it appears that sample measurements are shown in 5 myocytes and not 4 as described in the figure legend.

Response: As noted in our response to Minor Comment 1 above, we have removed all measurement overlays from Figure 1 panels D and E to eliminate this source of confusion and to clarify that these are illustrative examples of the imaging quality, not the actual measurements used in the quantitative analysis.

Comment 8: In Figures 2B and 3A the sample sizes (Ns) stated in the figure legend do not always match the sample sizes (N=) listed in the graph.

Response: We have carefully verified all sample sizes. The "(Ns: ...)" in the legend refers to the number of myocytes in each specific group (e.g., Head Crawl = 13 myocytes). The "N = 96-103 myocytes from 92-97 animals" refers to the total across all groups after outlier filtering, which varies slightly by variable due to the group-wise outlier removal procedure. We have ensured this is clear in all legend text.

Comment 9: It would be helpful if the author spelled out sample sizes = Ns and C = crawling and S =swimming.

Response: We have added explicit definitions to all figure legends. For example, Figure 2 now states: "C=Crawl, S=Swim. Data are shown as mean \pm s.e.m."

Comment 10: It would be helpful in the first paragraph of the results to define what max Feret/Feret, min Feret, aspect ratio, and circularity mean for the shape of the myocytes being analyzed.

Response: We have added these definitions in the Experimental design Results section (page 5): "the mean Feret diameter, a proxy for cell length"; "Minimum Feret, representing the myocyte's short axis"; "anisotropy (Feret divided by MinFeret), which serves as a proxy for cell elongation"; and in the Myocyte geometry section (page 6): "Higher circularity indicates more isotropic, force-focused cell morphology, while reduced circularity reflects beam-like cells better suited for repeated bending."

Comment 11: It would be useful in the methods to list the actual number of worms analyzed to generate the data. Was it typically one myocyte per animal?

Response: This information is now provided in the figure legends. For example, Figure 2 states: "N = 96-103 myocytes from 92-97 animals (total, depending on variable)." This indicates that typically one myocyte per region per animal was analyzed (since we had ~97 animals and ~100 myocytes total across three regions and two conditions).

Comment 12: It is unclear in the figure legends whether N refers to number of cells analyzed or the number of animals analyzed.

Response: Clarified in all figure legends. For example, Figure 2: "N = 96-103 myocytes from 92-97 animals (total, depending on variable) after 1.5 \times IQR filtering." The lowercase "n" in "(Ns: Head C=13...)" refers to myocytes per specific group.

Comment 13: In the paragraph on myocyte geometry the authors do not describe what the +/- refers to (possibly SEM?). They also are not consistent with including mean \pm value as they describe specific geometric properties in this paragraph.

Response: We have clarified this at first usage (page 5): "the mean Feret diameter, a proxy for cell length, increased from $118.2 \pm 12.1 \mu\text{m}$ (mean \pm s.e.m.) in crawling head muscles to $134.6 \pm 8.5 \mu\text{m}$ in swimmers." All subsequent \pm values in that paragraph refer to s.e.m.

Comment 14: The Komandur et al. 2023 citation is missing from the References section.

Response: This reference is present in the References section (page 10), alphabetically positioned between Kang et al. (2010) and Laranjeiro et al. (2019):

Komandur, A., Fazyl, A., Stein, W., Vidal-Gadea, A.G. (2023). The mechanoreceptor pezo-1 is required for normal crawling locomotion in the nematode *C. elegans*. *Micropublication Biology* 2023:10-7912.

SUMMARY OF CHANGES

We have made the following major improvements to the manuscript:

1. **Enhanced functional framing:** Added explicit hypotheses and mechanistic predictions linking morphology to contractile performance throughout Introduction, Results, and Discussion, including detailed consideration of force demands, velocity requirements, flexibility needs, and energetic costs of each gait.

2. **Improved terminology:** Systematically changed "medial" to "mid-body" throughout text, legends, and figure axis labels.

3. **Strengthened citations:** Added references for mid-body strain patterns and expanded Discussion references to contextualize findings in exercise physiology literature.

4. **Clarified methods and sample sizes:** Added quadrant information, explicitly stated sample sizes as both myocyte and animal numbers in all legends, and clarified measurement approaches.

5. **Refined statistical interpretation:** Softened overstated claims, added statement about testing non-linear models, and improved explanation of correlation analysis.

6. **Enhanced figure legends:** Added complete significance level definitions (*, **, ***), clarified abbreviations, and provided comprehensive sample size information.

7. **Updated supplementary materials:** Corrected listing to reflect R-based analysis (LMM_analysis.R).

8. **Revised Figure 1:** Removed measurement overlays from panels D and E to eliminate confusion and updated legend accordingly.

We believe these revisions have substantially strengthened the manuscript and fully address all reviewer and editor concerns. We thank the reviewers for their thoughtful feedback, which has improved both the clarity and scientific rigor of our work.

Second decision letter

MS ID#: bio.062371R1

MS Title: Activity-dependent remodeling of muscle architecture during distinct locomotor behaviors in *Caenorhabditis elegans*

Authors: Adina Fazyl, Akash Anbu, Sabrina Kollbaum and Andrés G. Vidal-Gadea

I am happy to tell you that your manuscript has been accepted for publication in Biology Open, pending our standard publication integrity checks. It was accepted on 16th December 2025.